# High rate of partner violence during pregnancy in eastern Ethiopia: Findings from a facility-based study

Abdulbasit Musa[1,2]*, Catherine Chojenta[2], Deborah Loxton[2]

**1** College of Health and Medical Sciences, Haramaya University, Harar, Ethiopia, **2** Research Centre for Generational Health and Ageing, Faculty of Health and Medicine, University of Newcastle, Newcastle, Australia

☯ These authors contributed equally to this work.

* atinaf.musa2@gmail.com

## Abstract

### Introduction

Intimate partner violence during pregnancy can contribute to maternal mortality and morbidity by limiting women's ability to receive maternal health services including antenatal care and skilled delivery care. In Ethiopia, evidence regarding intimate partner violence during pregnancy is limited, and no previous studies have been conducted in the Harari region. Therefore, this study aimed to investigate the prevalence and associated factors of intimate partner violence during pregnancy among women who had given birth in public hospitals in Harari regional state, eastern Ethiopia.

### Methods

A hospital-based cross-sectional study was conducted from November 2018 to April 2019 among women who had given birth in public hospitals in Harari regional state, East Ethiopia. A systematic random sampling method was employed to select 648 participants. Data were collected using an interviewer-administered standardized questionnaire based on the World Health Organization Multi-Country Study on Women's Health and Domestic Violence against Women survey. Crude and adjusted odds ratios with respective confidence intervals were computed. Variables with a p-value of ≤0.05 were considered to have a significant association with intimate partner violence during pregnancy.

### Results

The prevalence of intimate partner violence during the most recent pregnancy was found to be 39.81%. Furthermore, the prevalence of physical, emotional and sexual violence were found to be 25.93%, 25.62% and 3.7%, respectively. Longer duration of marriage (adjusted odds ratio = 1.68, 95% confidence interval = 1.01–2.79), most recent pregnancy being unplanned (adjusted odds ratio = 1.55, 95% confidence interval = 1.03–2.34), experiencing controlling behaviour by a partner, (adjusted odds ratio = 2.23, 95% confidence interval = 1.46–3.40) and having an attitude that justifies intimate partner violence (adjusted odds

**Data Availability Statement:** All relevant data are within the manuscript.

**Funding:** We received financial support as part of regular support for research student from the University of Newcastle, Australia, to support some

of our data collection and travel fees. The funder had no role in study design, data collection, and analysis, decision to publish, or preparation of the manuscript.

**Competing interests:** The authors have declared that no competing interests exist.

ratio = 1.60, 95% confidence interval = 1.09–2.36) were associated with experiencing intimate partner violence.

## Conclusion

The prevalence of intimate partner violence during pregnancy was found to be high. Pregnancy monitoring programs, which can detect and intervene with regard to partner's controlling behaviors and women's perception regarding justification of intimate partner violence, especially in those women with an unplanned pregnancy, could help to reduce intimate partner violence during pregnancy. Further, changing social norms that condone violence through advocacy and awareness creation might help in preventing partner violence.

## Introduction

In the current paper, we adopted the World Health Organisation's (WHO) definition that refers to intimate partner violence (IPV) as any physical, sexual, or psychological harm that is perpetrated by a current or former partner. It includes an intentional use of physical force (physical violence), humiliation, coercion, and intimidation (psychological violence), forced intercourse and other forms of sexual coercion (sexual violence) [1].

IPV during pregnancy has a detrimental effect on both the health of the mother and the newborn, including vaginal bleeding, [2] low birth weight [3], and preterm labor [2]. Furthermore, IPV can indirectly contribute to maternal mortality and morbidity through limiting a woman's ability to receive maternal health services like antenatal care [4–6], skilled delivery care [5, 6], and family planning services [7], which are effective interventions in reducing maternal mortality and morbidity. In addition to it contributing to 38% of murders of women worldwide [8], IPV also exposes women to mental health problems, including stress, anxiety, and depression [9].

Worldwide, the rates of violence perpetrated by intimate male partners during pregnancy varies from country to country. In a multi-country study, the WHO reported the prevalence of IPV during pregnancy ranged from 1% in Japan to 28% in Peru, with an average prevalence of 4–12% [10]. Findings of a systematic review of African studies reported a higher prevalence of IPV during pregnancy, ranging from 2% to 57% [11].

In Ethiopia, evidence regarding IPV during pregnancy is scarce, and to date, no studies have been conducted in the Harari region. The 2016 Ethiopian Demographic and Health Survey (EDHS) report has indicated a higher prevalence of IPV (37%) than the national prevalence (34%) among the general population in the Harari region [12]. As there were no previous studies that assessed IPV during pregnancy in the region, it is not known if this high prevalence of IPV among the general population is true for pregnant women.

The few available studies from other regions of Ethiopia have indicated IPV during pregnancy ranged from 12% in southern Ethiopia [13] to 75.2% in central Ethiopia [5]. Differences in forms of IPV studied can be a reason for the discrepancy as some of the studies only assessed physical violence [13, 14], while others examined physical, emotional and sexual IPV [5, 15–18]. Furthermore, a systematic review and meta-analysis of eight studies conducted in Ethiopia reported that 26.1% of women had experienced IPV during pregnancy [19]. However, the systematic review used studies that were conducted in only four regions of Ethiopia's nine regions and none of the studies were from the Harari region [19].

Harari region has the highest proportion of population who reside in urban (55%) than any other regions in Ethiopia and the majority of its population are Muslim (69%) [20]. Both residency [21] and religion [22] are important factors that can affect the prevalence of IPV. For example, evidence indicated that in some Muslim countries, selective quotes from the Quran may be used to 'prove' that it is allowable for a man to 'beat' his wife [22] indicating the role of religion in promoting social control. Therefore, the generalizability of the study that was conducted in one part of the country might not provide the true prevalence of IPV in the other region as IPV might be affected by within-country differences in cultures and societal norms [23] since Ethiopia is a multi-cultural state [24].

The relationship between dowry payment and partner violence has long been established [25]., Unlike countries in Asia that offer a dowry to the groom's family [26], in Ethiopia a dowry of cash or cattle is given to the bride's family before the marriage [27]; this is practiced in Harari region. A previous Ethiopian study indicated that women who considered that dowry had an impact on the way she was treated by her husband and his family were more likely to experience IPV during pregnancy than those who considered it had no impact on the way she was treated [16]. Other researchers from Ethiopia also reported associations between IPV during pregnancy and a woman and her partner being uneducated [19], early marriage, partner's alcohol consumption [14], chewing khat, [13], and rural residency [17]. Other factors that have been found to have an association with IPV include controlling behaviour by the partner [28], duration of marriage [29], and engaging in polygamous relationships [30]. However, these have not been comprehensively studied in Ethiopia, indicating the importance of conducting further studies to understand these factors and generate evidence that will enable the design of strategies that address IPV.

Therefore, this study aimed to investigate the prevalence and associated factors of IPV during pregnancy among women who gave birth in public hospitals in the Harari region, East Ethiopia. In other words, the study attempted to address the following question: is there an association between IPV and key sociodemographic factors, marital and behavioural factors, reproductive and physical health-related factors, attitudes towards IPV and partner controlling behaviour?

## Methods

### Setting

The study was conducted in public hospitals found in Harari regional state, East Ethiopia. Harari region is one of the nine regions in Ethiopia. Harar, the capital city of the region, is 526 km east of Addis Ababa, the capital city of Ethiopia. According to the 2013 Ethiopian Central Statistical Agency's population projection, the region has a population of 246,000 of whom 137,000 (55.7%) are urban residents. Annually, over 7,000 women are expected to be pregnant in the region [20]. According to the Regional Health Bureau, the health service coverage of the region is 100%, with institutional delivery coverage at 95%. There are two public hospitals, the Federal Police Hospital, two private general hospitals, one fistula hospital, eight government health centers, 16 health posts, and one non-governmental organization clinic. The study was conducted in the two public hospitals: Hiwot Fana Specialized University Hospital and Jugal Hospital.

### Study design

A cross-sectional survey of women who had given birth in public hospitals found in the Harari region was conducted from 25 November 2018 to 28 April 2019.

## Participants

All women who gave birth in the public hospitals found in the Harari region were included in the eligible study population. Participants were recruited following the birth of their baby in the hospitals. Using a previously developed and published protocol [31], we intended to exclude women with a high risk for emotional distress and in imminent danger. However, no women at such risk were identified during data collection.

The sample size was calculated using significant factors of partner violence during pregnancy obtained from a previous study conducted in Ethiopia [16]. Dowry payment maximized sample size; therefore, we used it as exposure variable for calculating sample size considering the following parameter: dowry payment as the exposure variable, power level of 80%, and the ratio of unexposed to exposed of 1.0. Therefore, taking a 5% contingency for the non-response rate, the final sample size needed was 670.

To select study participants, a systematic random sampling method was used. According to information from the hospitals, on average 2,283 women give birth in five months in the selected hospitals. Of these women, 1,599 give birth in Hiwot Fana Specialized University Hospital while the rest (684) give birth at Jugal Hospital. Therefore, proportionally, we recruited 469 participants from Hiwot Fana Specialized University Hospital while the remaining 201 were recruited from Jugal Hospital, using a sampling interval of 3 for both Hiwot Fana Specialized University Hospital (1599/469 = 3.41) and Jugal hospital (610/201 = 3.40).

## Study tool and measurements

We used the WHO Multi-Country Study on Women's Health and Domestic Violence Against Women questionnaire [32] to assess partner violence during the most recent pregnancy. The WHO questionnaire is available for public use and it was previously adapted to the Ethiopian context [32]. In this study, a woman was considered to have experienced IPV during pregnancy if she reported experiencing any violence during the most recent pregnancy including physical, sexual, or emotional violence. Physical, psychological and sexual IPV were measured by 6, 4 and 3 items respectively (Table 1). All items had response options of 'yes' or 'no'.

**Table 1. Items used to measure different types of IPV in the current study.**

| IPV Types | | Items asked | Yes | No |
|---|---|---|---|---|
| Physical | During your most recent pregnancy, did your partner: | Slap you or throw something at you that could hurt you? | | |
| | | Push you, shove you or pull your hair | | |
| | | Hit you with his fist or with something else that could hurt you? | | |
| | | Kick you, drag you about or beat you up? | | |
| | | Attempt to choke you or burn you on purpose? | | |
| | | Threaten to use or actually use a gun, knife or other weapon against you? | | |
| Emotional | During your most recent pregnancy, did your partner: | Say or do something to humiliate you in front of another person? | | |
| | | Threaten to hurt or harm you or someone you care about? | | |
| | | Insult you or make you feel bad about yourself? | | |
| | | Do things to scare or intimidate you on purpose (e.g., by the way he looked at you, by yelling and smashing things)? | | |
| Sexual | During your most recent pregnancy, did your partner | Physically force you to have sexual intercourse with him when you did not want to? | | |
| | | Force you with threats or in any other way to perform sexual acts you did not want? | | |
| | | Physically force you to perform any other sexual acts you did not want to? | | |

Women who responded 'yes' to at least one of physical, emotional, or psychological IPV item were considered to have experienced IPV.

In addition, potential explanatory variables identified by reviewing literature included the following sociodemographic factors: information regarding both partners' age, education, and occupation, residency, region, religion, ethnicity, monthly family income, and the availability of social support for women. Marital and behavioural factors: information regarding the marital status of women, age at first marriage, number of lifetime sexual partners of women, duration of marriage, marriage ceremony, women's ability to choose their partner, dowry payment during marriage, polygamous union, a woman and her partner's khat chewing habit, and smoking and drinking alcohol during the most recent pregnancy. Reproductive and physical health-related factors: gravidity, parity, maternal pregnancy plan, partner's pregnancy plan, partner's child sex preference, women's HIV serostatus, women ever diagnosed with kidney disease, and individual and family history of mental health problems.

Social support was measured using the Multidimensional Scale of Perceived Social Support (MDSPSS) questionnaire [33], which contains 12 questions, each followed by seven possible responses in a Likert scale format ranging from 'very strongly disagree' to 'very strongly agree'. Women scoring the mean and above in the MDSPSS questionnaire were considered to have good support, while those who scored below the mean were considered to have poor support [33].

Attitude regarding IPV was evaluated by asking the participants if they agreed that a husband can beat his wife under each of the following six conditions: if she doesn't complete her household work to his satisfaction; if she disobeys him; if she refuses to have sex; if she asks him whether he has another girlfriend; if he suspects that she is unfaithful; and if he found out that she is unfaithful. Participants who agreed to at least one of the above conditions were considered to have a perception that might justify IPV.

Similarly, women were also asked about their partner's controlling behavior. These items included: whether he is jealous/angry if she talks to another male; if he has accused her of being unfaithful; if he does not permit her to meet female friends; if he has tried to limit her contact with her family; and if he insists on knowing where she is at all time. Women who responded to experiencing at least one of the above conditions with their current partner were considered to have experienced controlling behavior by a partner.

Women's attitude regarding ability to refuse sex was assessed by asking women whether they believed that women have the right to refuse sex with their partners under the following conditions: if she does not want to have sex; if he has mistreated her; if she is sick; and if he is drunk. Respondents who agreed that women have the right to refuse sex under any of these circumstances were considered to have a positive attitude towards women's right to refuse sex.

The prepared study tool was pretested on women from a similar facility which was not selected for this study to ensure consistency in providing the required information. Trained female midwives collected data using a face-to-face interviewer-administered questionnaire. The supervisors carefully followed the data collection procedure on a daily basis to ensure smooth data collection throughout the data collection period.

### Statistical analysis

Data were entered into Epi Data version 3.1 and exported to STATA version 14 software package for analysis. The relationship between possible explanatory variables and IPV was investigated by using logistic regression. Three logistic regression models were fitted to investigate the associated factors of IPV. In the first model (Model 1), binary logistic regression was conducted to determine associations between IPV and potential explanatory variables. All

potential explanatory variables that were fitted in binary logistic regression were obtained from relevant literatures. In the second model (Model 2), variables (excluding attitude towards IPV and partner controlling behaviour) that demonstrated a significant association (with a p-value of ≤0.05) in bivariate analysis were exported to a multiple logistic regression model for multivariate analysis. In the final model (Model 3), the attitude towards IPV and controlling behaviour were added to variables in Model 2 to investigate the relationship between IPV and all explanatory variables. Both crude and adjusted odds ratios with 95% confidence intervals were calculated when fitting logistic regression model. Variables with a p-value of ≤0.05 in multivariate analysis were considered to have a significant association with IPV during pregnancy.

### Ethical clearance

Ethical clearance was obtained from the Human Research Ethics Committee of the University of Newcastle (approval number: H-2018-0160), and the Institutional Research Ethics Review Committee of Haramaya University (approval number: IHRERC/195/2018). Permission to conduct the study was sought from each hospital administration. Voluntary verbal informed consent was obtained from each study participant. First, the interviewer read out participant's information sheet slowly. Information regarding purpose of the study, risk and benefit of participating, measure taken to protect participants' privacy and their choice to participate in the study were explained. Then, they were given adequate time to ask questions to ensure they had a clear understanding of the study. After their questions were addressed, their consent was sought by asking the participant to respond affirmatively to a series of statements. The interviewer signed and dated the consent form. The privacy of the participants was assured during the interview and they were not interviewed in the presence of their male partner to reduce the risk to participants and the interviewer. The interviews adhered to WHO Ethical and Safety recommendations for research on domestic violence against women [34]. All participants who reported experiencing IPV were asked if they needed assistance; however, none of them took up the offer to be linked to the local women and children's affairs office for support (the office responsible for the safety of women in Ethiopia).

## Results

### Socio-demographic characteristics of the respondents

Among the potential 670 respondents, 648 women took part in the study, making a response rate of 96.7%. The majority of the respondents (83.18%) were aged 20–34 years, while most of their partners (78.55%) were in the age group of 20–39 years. Almost half of the women (48.61%) and a third of their partners (31.02%) had never attended any formal education. Most of the women (79.78%) were not involved in paid work while about half of the husbands (50.46%) were farmers/daily labourer. More than half of the respondents (55.40%) reported having good social support (Table 2).

### Marital and behavioral factors of the respondents

Among all respondents, 99.07% were married and approximately half of them (49.69%) had lived with their current partner for more than five years. Almost all of the respondents, 96.60% were married in a ceremony, and most of the women (93.52%) chose their partner. About half of the respondents (49.23%) and most of their partners (84.88%) chewed khat during the most recent pregnancy (Table 3).

**Table 2. Socio-demographic characteristics of women who gave birth in public hospitals in Harari regional state, East Ethiopia, 2018–2019.**

| Variables | N | % | IPV | | | | P-value |
|---|---|---|---|---|---|---|---|
| | | | Yes | | No | | |
| | | | N | % | N | % | |
| Woman's age | | | | | | | |
| <20 years | 29 | 4.48 | 11 | 37.93 | 18 | 62.07 | 0.006 |
| 20–34 | 539 | 83.18 | 202 | 37.48 | 337 | 62.52 | |
| 35 and above | 80 | 12.34 | 45 | 56.25 | 35 | 43.75 | |
| Partners' age | | | | | | | |
| 20–39 | 509 | 78.55 | 183 | 35.95 | 326 | 64.05 | <0.001 |
| 40 and above | 139 | 21.45 | 75 | 53.96 | 64 | 46.04 | |
| Residence | | | | | | | |
| Rural | 248 | 38.27 | 99 | 39.92 | 149 | 60.08 | 0.966 |
| Urban | 400 | 61.73 | 159 | 39.75 | 241 | 60.25 | |
| Changed residential area during the most recent pregnancy | | | | | | | |
| Yes | 81 | 12.50 | 37 | 45.68 | 44 | 54.32 | 0.249 |
| No | 567 | 87.50 | 221 | 38.98 | 346 | 61.02 | |
| Region | | | | | | | |
| Oromia | 388 | 59.88 | 165 | 42.53 | 223 | 57.47 | * |
| Harari | 256 | 39.51 | 90 | 35.16 | 166 | 64.84 | |
| Somali | 4 | 0.62 | 3 | 75.00 | 1 | 25.00 | |
| Religion | | | | | | | |
| Muslim | 535 | 82.56 | 221 | 41.31 | 314 | 58.69 | 0.091 |
| Christian | 113 | 17.44 | 37 | 32.74 | 76 | 67.26 | |
| Woman's education | | | | | | | |
| No formal education | 315 | 48.61 | 149 | 47.30 | 166 | 52.7 | <0.001 |
| Formal education | 333 | 51.39 | 109 | 32.73 | 224 | 67.27 | |
| Partners' education | | | | | | | |
| No formal education | 201 | 31.02 | 100 | 49.75 | 101 | 50.25 | <0.001 |
| Primary education | 447 | 68.98 | 158 | 35.35 | 289 | 64.65 | |
| Woman's occupation | | | | | | | |
| Not paid work | 517 | 79.78 | 215 | 41.59 | 302 | 58.41 | 0.067 |
| Paid work | 131 | 20.22 | 43 | 32.82 | 88 | 67.18 | |
| Partners' occupation | | | | | | | |
| Farmer/daily labourer | 327 | 50.46 | 105 | 32.11 | 222 | 67.89 | 0.581 |
| Self-Employed | 191 | 29.48 | 65 | 34.03 | 126 | 65.97 | |
| Other(NGO/students) | 130 | 20.06 | 39 | 30.00 | 91 | 70.16 | |
| Average monthly family income in Ethiopian Birr | | | | | | | |
| 400–1000 | 97 | 14.97 | 47 | 48.45 | 50 | 51.55 | 0.023 |
| 1001–2000 | 201 | 31.02 | 88 | 43.78 | 113 | 56.22 | |
| 2001 and above | 350 | 54.01 | 123 | 35.14 | 227 | 64.86 | |
| Received financial support | | | | | | | |
| Yes | 21 | 3.24 | 11 | 52.38 | 10 | 47.62 | 0.232 |
| No | 627 | 96.76 | 247 | 39.39 | 380 | 60.61 | |
| Social support | | | | | | | |
| Good support | 359 | 55.40 | 131 | 36.49 | 228 | 63.51 | 0.054 |
| Poor support | 289 | 44.60 | 127 | 43.94 | 162 | 56.06 | |

*chi2 p-value was not calculated due to small cell count.

**Table 3. Marital and behavioral factors of women who gave birth in public hospitals in Harari regional state, East Ethiopia, 2018–2019.**

| Variables | N | % | IPV | | | | P-value |
|---|---|---|---|---|---|---|---|
| | | | Yes | | No | | |
| | | | N | % | N | % | |
| Marital status | | | | | | | |
| Married | 642 | 99.07 | 252 | 39.25 | 390 | 60.75 | * |
| Others | 6 | 0.93 | 6 | 100.00 | - | - | |
| Woman's lifetime sexual partners | | | | | | | |
| Single | 600 | 92.59 | 233 | 38.83 | 367 | 61.17 | 0.071 |
| Multiple | 48 | 7.41 | 25 | 52.08 | 23 | 47.92 | |
| Duration of marriage to current partner | | | | | | | |
| Under five years | 326 | 50.31 | 104 | 31.90 | 222 | 68.10 | <0.001 |
| More than five years | 322 | 49.69 | 154 | 47.83 | 168 | 52.17 | |
| Age at first marriage | | | | | | | |
| Under 18 years | 116 | 17.90 | 49 | 42.24 | 67 | 57.76 | 0.556 |
| 18 years and over | 532 | 82.10 | 209 | 39.29 | 323 | 60.71 | |
| Marriage ceremony | | | | | | | |
| Yes | 626 | 96.60 | 249 | 39.78 | 377 | 60.22 | 0.915 |
| No | 22 | 3.40 | 9 | 40.91 | 13 | 59.09 | |
| Choice of partner | | | | | | | |
| The woman | 606 | 93.52 | 242 | 39.93 | 364 | 60.07 | 0.814 |
| Others | 42 | 6.48 | 16 | 38.10 | 26 | 61.90 | |
| Dowry requested | | | | | | | |
| Yes | 188 | 29.01 | 77 | 40.96 | 111 | 59.04 | 0.704 |
| No | 460 | 70.99 | 181 | 39.35 | 279 | 60.65 | |
| Dowry payment before marriage (n = 188) | | | | | | | |
| Fully Paid | 174 | 92.55 | 72 | 41.38 | 102 | 58.62 | 0.678 |
| Partially paid | 14 | 7.45 | 5 | 35.71 | 9 | 64.29 | |
| Perceived dowry impact on relationship (n = 188) | | | | | | | |
| Positive impact | 25 | 13.30 | 11 | 44.00 | 14 | 56.00 | 0.160 |
| Negative | 18 | 9.57 | 11 | 61.11 | 7 | 38.89 | |
| No impact | 145 | 77.13 | 55 | 37.93 | 90 | 62.07 | |
| Living with partner's relative | | | | | | | |
| Yes | 212 | 32.72 | 76 | 35.85 | 136 | 64.15 | 0.150 |
| No | 436 | 67.29 | 182 | 41.74 | 254 | 58.26 | |
| Suspects partner of infidelity | | | | | | | |
| Yes | 12 | 1.85 | 7 | 58.33 | 5 | 41.67 | * |
| No | 636 | 98.15 | 251 | 39.47 | 385 | 60.53 | |
| Polygamous marriage | | | | | | | |
| Yes | 22 | 3.40 | 10 | 45.45 | 12 | 54.55 | 0.582 |
| No | 626 | 96.61 | 248 | 39.62 | 378 | 60.38 | |
| Position of the respondent in a polygamous union (n = 22) | | | | | | | |
| First wife | 4 | 18.18 | 1 | 25.00 | 3 | 75.00 | * |
| Second wife | 17 | 77.27 | 8 | 47.06 | 9 | 52.94 | |
| Third wife | 1 | 4.55 | 1 | 100.00 | - | - | |
| Change in sexual interest during the last pregnancy | | | | | | | |
| No change | 378 | 58.33 | 162 | 42.86 | 216 | 57.14 | * |
| Reduced | 268 | 41.36 | 94 | 35.07 | 174 | 64.93 | |
| Increased | 2 | 0.31 | 2 | 100.00 | - | - | |

(*Continued*)

**Table 3.** (Continued)

| Variables | N | % | IPV | | | | P-value |
|---|---|---|---|---|---|---|---|
| | | | Yes | | No | | |
| | | | N | % | N | % | |
| Maternal khat chewing during last pregnancy | | | | | | | |
| Ever chewed | 319 | 49.23 | 105 | 32.92 | 214 | 67.08 | 0.641 |
| Never chewed | 329 | 50.77 | 114 | 34.65 | 215 | 65.35 | |
| Maternal cigarette smoking during last pregnancy | | | | | | | |
| Ever smoked | 4 | 0.62 | 3 | 75.00 | 1 | 25.00 | * |
| Never smoked | 644 | 99.38 | 255 | 39.60 | 389 | 60.40 | |
| Maternal alcohol use during last pregnancy | | | | | | | |
| Ever drunk | 56 | 8.64 | 19 | 33.93 | 37 | 66.07 | 0.346 |
| Never drunk | 592 | 91.36 | 239 | 40.37 | 353 | 59.63 | |
| Partner's khat chewing during last pregnancy | | | | | | | |
| Ever chewed | 550 | 84.88 | 193 | 35.10 | 357 | 64.90 | 0.390 |
| Never chewed | 98 | 15.12 | 30 | 30.61 | 68 | 69.39 | |
| Partner's smoking during last pregnancy | | | | | | | |
| Ever smoked | 142 | 21.91 | 70 | 49.30 | 72 | 50.70 | 0.009 |
| Never smoked | 506 | 78.09 | 188 | 37.15 | 318 | 62.85 | |
| Partner's alcohol use during last pregnancy | | | | | | | |
| Ever drunk | 74 | 11.42 | 25 | 33.78 | 49 | 66.22 | 0.260 |
| Never drunk | 574 | 88.58 | 233 | 40.59 | 341 | 59.41 | |

*chi2 p-value was not calculated due to small cell count.

### Reproductive and physical health-related factors of the respondents

Among all respondents, one-third (33.3%) were primigravida, whereas 37.04% of them were primipara. Of all respondents, more than three quarters (76.54%) reported that they had planned their most recent pregnancy (Table 4).

### Attitudes supportive of partner violence and partner controlling

The findings indicated that almost all respondents (99.38%) agreed that married women should be able to refuse sex with her husband in certain circumstances. More than two-thirds of the respondents (68.67%) had a perception that might justify IPV (Fig 1).

### Intimate partner violence

Women were asked about their experience regarding exposure to different types of violence from their intimate partners. Slapping/throwing things at them (24.69%) and making the women feel bad about themselves (17.75%) constituted the most frequently experienced emotional and physical violence, respectively. Forcing the women to have sex when they did not want to (2.31%) was found to be the most frequently reported sexual violence (Fig 2).

Of all respondents, 39.81% reported experiencing any of the three aspects of IPV, while the least frequently reported types of IPV being sexual violence, 3.70% (Table 5).

### Factors associated with partner violence during most recent pregnancy

The relationship between all potential explanatory variables and IPV were investigated by using binary logistic regression (Model 1; Table 6). In the second model (Model 2), variables

**Table 4. Reproductive and physical health-related factors of women who gave birth in public hospitals in Harari regional state, East Ethiopia, 2018–2019.**

| Variables | N | % | IPV | | | | P-value |
|---|---|---|---|---|---|---|---|
| | | | Yes | | No | | |
| | | | n | % | N | % | |
| Gravidity | | | | | | | |
| Primigravida | 216 | 33.33 | 69 | 31.94 | 147 | 68.06 | 0.004 |
| Multigravida | 432 | 66.67 | 189 | 43.75 | 243 | 56.25 | |
| Parity | | | | | | | |
| Primipara | 240 | 37.04 | 76 | 31.67 | 164 | 68.33 | 0.001 |
| Multipara | 408 | 62.96 | 182 | 44.61 | 226 | 55.39 | |
| Maternal pregnancy plan | | | | | | | |
| Planned | 496 | 76.54 | 177 | 35.69 | 319 | 64.31 | <0.001 |
| Unplanned | 152 | 23.46 | 81 | 53.29 | 71 | 46.71 | |
| Maternal efforts in delaying their last pregnancy | | | | | | | |
| Yes | 25 | 3.86 | 12 | 48.00 | 13 | 52.00 | 0.394 |
| No | 623 | 96.14 | 246 | 39.49 | 377 | 60.51 | |
| ANC Booking | | | | | | | |
| Early | 328 | 50.62 | 145 | 44.21 | 183 | 55.79 | 0.215 |
| Late/No | 320 | 49.38 | 157 | 49.06 | 163 | 50.94 | |
| Partner's preference of child's sex | | | | | | | |
| Son | 110 | 16.98 | 40 | 36.36 | 70 | 63.64 | 0.545 |
| Daughter | 53 | 8.18 | 24 | 45.28 | 29 | 54.72 | |
| No preference | 485 | 74.84 | 194 | 40.00 | 291 | 60.00 | |
| Women's current HIV status | | | | | | | |
| Positive | 2 | 0.31 | 1 | 50.00 | 1 | 50.00 | * |
| Negative | 512 | 79.01 | 180 | 35.16 | 332 | 64.84 | |
| Unknown | 134 | 20.68 | 77 | 57.46 | 57 | 42.54 | |
| Diagnosed with kidney disease | | | | | | | |
| Yes | 99 | 15.28 | 44 | 44.44 | 55 | 55.56 | 0.307 |
| No | 549 | 84.72 | 214 | 38.98 | 335 | 61.02 | |
| Has a known disability | | | | | | | |
| Yes | 5 | 0.77 | 3 | 60.00 | 2 | 40.00 | * |
| No | 643 | 99.23 | 255 | 39.66 | 388 | 60.34 | |
| Diagnosed with mental illness | | | | | | | |
| Yes | 8 | 1.23 | 7 | 87.50 | 1 | 12.50 | * |
| No | 640 | 98.77 | 251 | 39.22 | 389 | 60.78 | |
| Family member diagnosed with a mental illness | | | | | | | |
| Yes | 16 | 2.47 | 10 | 62.50 | 6 | 37.50 | * |
| No | 632 | 97.53 | 248 | 39.24 | 384 | 60.70 | |

*chi2 p-value was not calculated due to small cell count.

that demonstrated a significant association (with a p-value of ≤0.05) in bivariate analysis (excluding attitude towards IPV and partner controlling behaviour) were exported to a multiple logistic regression model for multivariate analysis. In the final model (Model 3), the attitude towards IPV and controlling behaviour were added to variables in Model 2 to investigate the relationship between IPV and all explanatory variables. As observed from Model 3, after controlling for possible confounders, duration of marriage with current partner, maternal pregnancy plan, partner controlling behavior and having a perception whereby IPV is justified

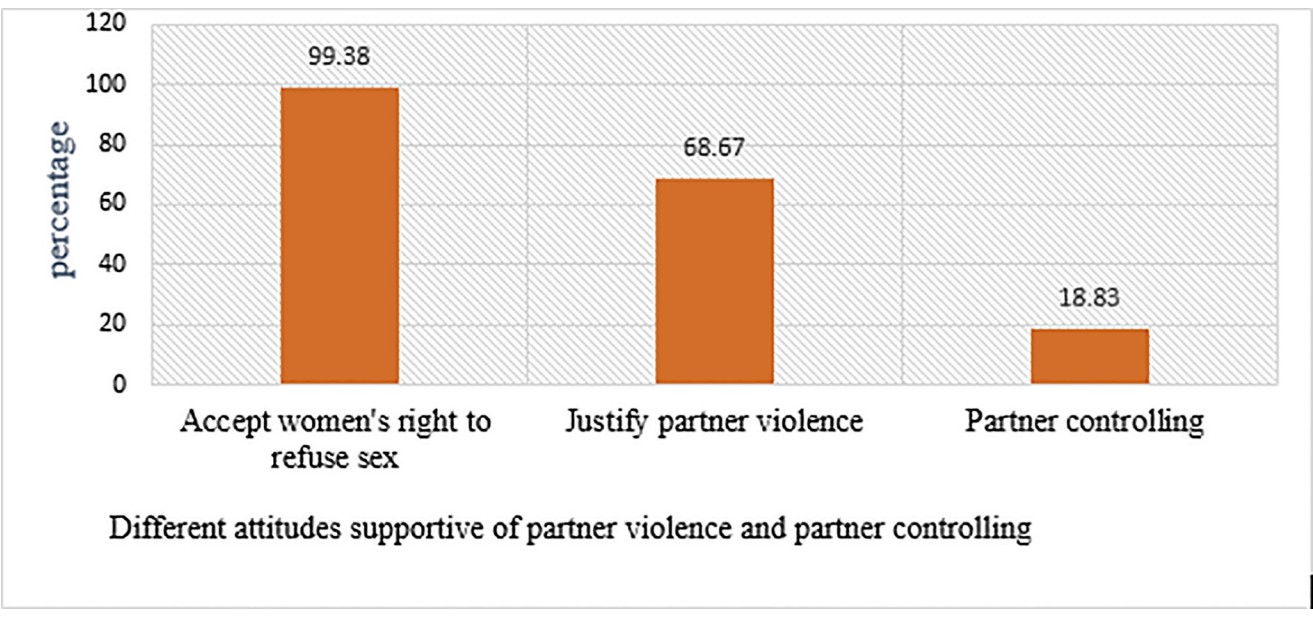

**Fig 1. Proportion of respondents having different perception that might justify partner violence and partner controlling among women who had given birth in public hospitals in Harari regional state, East Ethiopia, 2018–2019.**

were found to have an association with IPV in multivariate analysis. The study demonstrated that women with longer (>5 years) marriages with their current partner were 1.7 times (AOR = 1.68, 95%CI = 1.01–2.79) more likely to experience IPV. Furthermore, women who reported having an unplanned recent pregnancy were 1.6 times more likely (AOR = 1.55, 95% CI = 1.03–2.34) to experience IPV compared to those who had planned their pregnancy.

Women who reported partner controlling behavior and those who had a perception whereby IPV was justified were 2.2 (AOR = 2.23, 95%CI = 1.46–3.40) and 1.6 (AOR = 1.60,

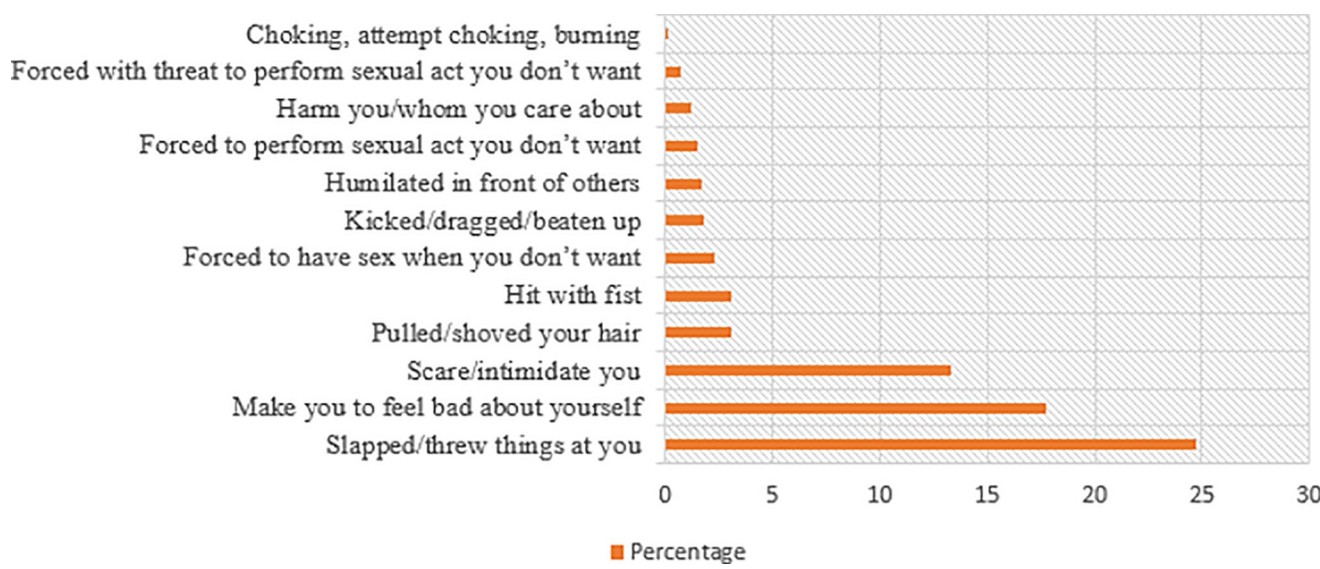

**Fig 2. Proportion of women who reported different types of IPV during pregnancy among women who had given birth in public hospitals in Harari regional state, East Ethiopia, 2018–2019.**

**Table 5. Proportion of women who reported different aspects of IPV during pregnancy among women who had given birth in public hospitals in Harari regional state, East Ethiopia, 2018–2019.**

| Types of partner violence reported | N | % | 95%CI |
|---|---|---|---|
| Emotional IPV | 166 | 25.62 | 22.3–29.16 |
| Physical IPV | 168 | 25.93 | 22.59–29.48 |
| Sexual IPV | 24 | 3.70 | 2.39–5.46 |
| Any IPV | 258 | 39.81 | 36.02–43.70 |

**Table 6. Bivariate and multivariate regression of factors associated with partner violence during pregnancy among women who gave birth in public hospitals in Harari regional state, East Ethiopia, 2018–2019.**

| Variable | Model 1 COR, 95%CI | Model 2 AOR, 95%CI | Model 3 AOR, 95%CI |
|---|---|---|---|
| Women age | | | |
| <20 years | 0.48(0.20, 1.14) | 1.04(0.37, 2.89) | 0.86(0.30, 2.47) |
| 20–34 years | 0.47(0.29, 0.75) | 0.78(0.42, 1.44) | 0.77(0.41, 1.44) |
| 35 years and over | 1 | 1 | 1 |
| Partners' age | | | |
| 20–39 years | 1 | 1 | 1 |
| 40 years and over | 2.09(1.43, 3.05) | 1.18(0.70, 1.98) | 1.22(0.72,2.08) |
| Women's education | | | |
| No formal education | 1.85(1.34, 2.54) | 1.19(0.78, 1.81) | 1.09(0.71, 1.68) |
| Has formal education | 1 | 1 | **1** |
| Partners' education | | | |
| No formal education | 1.81(1.29, 2.54) | 1.12(0.72, 1.71) | 1.12(0.72, 1.75) |
| Has formal education | 1 | | 1 |
| Average family income (in Eth Birr) | | | |
| 400–1000 | 1.74(1.10, 2.73) | 1.46(0.89, 2.40) | 1.63(0.98, 2.71) |
| 1001–2000 | 1.44(1.01, 2.05) | 1.27(0.87, 1.86) | 1.34(0.91, 1.98) |
| 2000 and above | 1 | 1 | 1 |
| Duration of marriage with current partner | | | |
| ≤5 years | 1 | 1 | 1 |
| >5years | 1.96(1.42, 2.69) | 1.53(0.94, 2.51) | **1.70(1.02, 2.82)*** |
| Parity | | | |
| Primi | 1 | 1 | 1 |
| Multi | 1.74(1.24, 2.43) | 0.99(0.60, 1.64) | 0.94(0.56, 1.57) |
| Husband ever smoked | | | |
| Yes | 1.65(1.13, 2.39) | 1.18(0.78, 1.78) | 1.02(0.67, 1.55) |
| No | 1 | 1 | 1 |
| Maternal pregnancy plan | | | |
| Planned | 1 | 1 | 1 |
| Unplanned | 2.06(1.42, 2.97) | 1.51(1.01, 2.25) | **1.58(1.06, 2.38)*** |
| Partner controlling behaviour | | | |
| Yes | 2.14(1.43, 3.18) | | **2.27(1.49, 3.46)*** |
| No | 1 | | 1 |
| Attitude towards partner violence | | | |
| Have perception that justified violence | 1.85(1.30, 2.63) | | **1.62(1.10 2.39)*** |
| Have no perception that justify violence | 1 | | 1 |

* Significant at a p-value of ≤0.05, COR = Crude odds ratio, AOR = Adjusted odds ratio, CI Confidence interval.

95%CI = 1.09–2.36) times more likely to experience IPV, respectively, compared to their counterparts (Table 6).

## Discussion

This study assessed the prevalence and associated factors of IPV during pregnancy in the Harari regional state, East Ethiopia. Overall, 40% of women reported experiencing IPV during their most recent pregnancy. Duration of marriage with current partner, maternal pregnancy plan, partner controlling behavior and having an attitude that justifies partner violence were found to have an association with IPV. The findings of the current study provide, for the first time, information regarding IPV during pregnancy in the region.

The prevalence of IPV in this study was similar to that of previous Ethiopian studies that were conducted in Jimma (45%) [16] and Tigray (41%) [35] and findings from other countries that were conducted in Jordan (40.9%) [36], Kenya (37%) [30], Egypt (44.1%) [37] and Portugal (43.4%) [38]. However, it was lower than that of previous studies from Ethiopia that were conducted in Addis Ababa (75.2%) [5] and Bale Zone (59%) [18], but higher than that of previously conducted research in southern Ethiopia (21%) [39]. The lower prevalence of IPV in the current study compared to the previous study from Bale Zone might be due to the difference in measurement of IPV as the study from Bale Zone used a wider definition of IPV that included economic violence, whereas in our study the definition of IPV was limited to physical, sexual, and emotional violence. Furthermore, the difference in the study setting might contribute to the lower prevalence of IPV in the current study compared to a previous community-based study from Addis Ababa. As previous evidence has indicated [6], women who attend health facilities for maternal health care services were less likely to report IPV compared to women in the community who did not attend the facility which might lead to the lower prevalence of IPV in the current study.

Differences in culture, social norms and implementation of laws that prevent violence against women might also lead to differences in prevalence of IPV [40]. For example, evidence from the 2016 EDHS has indicated a lower proportion of males support wife beating in southern Ethiopia (14.9%) compared to the Harari region (22.6%)[12] suggesting that at least some differences in social norms exist that might contribute to the differences in prevalence. Furthermore, the difference in gender equality might also contribute to the difference in prevalence of IPV as previous evidence has indicated a lower rate of partner violence in a society that upholds gender equality [41].

In this study, sexual violence (3.70%) was the least frequently reported type of IPV compared to both physical (25.93%) and emotional (25.62%) IPV. This finding is similar to the previous study conducted in Gondar Hospital, which reported a prevalence of sexual IPV of 2.4% [42]. However, a higher prevalence of sexual IPV among pregnant women was reported by other studies from Ethiopia that were conducted in Jimma (30%) [16], Debre-Berhan (19.8%) [17], Hulet Ejju Enessie district (14.8%) [43], and Bale Zone (36.3%) [18]. The low prevalence of sexual IPV in the current study might be attributed to women's strong disapproval of sexual violence as more than 99% of women in this study believed that women had the right to refuse sex. However, the potential for underreporting should not be overlooked as respondents might not give an answer honestly when asked culturally sensitive questions about sexual matters [44–46]. In sociocentric and highly patriarchal societies like Ethiopia, women often feel humiliated and ashamed to disclose sexual violence, primarily due to a lack of support and negative responses from others within their society because of the presence of cultural messages that trivialize sexual violence [47, 48].

The current study has indicated higher odds of IPV among women who were married for more than five years compared to those who were married for less than five years. A similar finding was reported by the study from Zambia indicating an increased risk of IPV as marriage duration increases [29]. The increased duration of marriage might lead to an increased risk of partner violence as factors such as having additional children and economic hardship put more demands and stress on the family[49].

Women with an unplanned pregnancy also had increased odds of experiencing partner violence during pregnancy. This finding is in agreement with previous studies from Ethiopia [13, 42, 43] and other review studies [50, 51]. This might indicate the existence and continuation of pre-pregnancy IPV because women in abusive relationships may not have control over when to have sex, and they might be less able to use effective methods of contraception [52]. Furthermore, it is possible that some men use intimidation and physical violence to pressure their partners to become pregnant [53], leading to unplanned pregnancy as an outcome of IPV.

Partner controlling behavior was also found to be associated with experiencing IPV. This finding is consistent with previous studies from Nigeria [28], Haiti [54] and Nepal [55]. This might be a result of the social construct that promotes male dominance through encouraging men to exercise control over their partner, which in turn paves the way for IPV when the male partner feels threatened by a perceived lack of control over their wives [56].

The study also demonstrated that women who had a perception whereby partner violence was justified were more likely to experience partner violence than those who had no such perception. This finding is consistent with previous studies conducted in Bangladesh [57] and Iran [58]. It is important to note that none of the women who experienced IPV in the current study took up the offer to be supported. Although further study is needed to understand why abused women may not always seek support, this might indicate an overall tolerance of violence within the community and its role in fostering a conducive environment for abusers [59]. Furthermore, the fact that women in Ethiopia have less control over financial resources, which includes reduced access to sources of money and ability to spend it without consulting their partners [60], might contribute to the acceptance of partner violence by reducing women's empowerment [61]. In countries like Ethiopia, where women have little control over financial resources but have a greater social responsibility for raising children, women may accept the violence in order to give priority to their children who might suffer if the family separated [62].

## Strengths and limitations

Due to its cross-sectional nature, the study cannot determine cause and effect relationships between IPV and other explanatory variables. Furthermore, there is a possibility of recall bias and social desirability bias, although we made efforts to reduce it by strictly adhering to the WHO Ethical and Safety Recommendations for Research on domestic violence against women [34]. Regardless of these limitations, the current study provides robust and much needed information on the scale of partner violence during pregnancy and its associated factors in public hospitals found in the region.

## Conclusion

The prevalence of IPV during pregnancy is high; two in five women experienced IPV during their most recent pregnancy. Pregnancy monitoring programs, which can detect and intervene with regard to partner's controlling behaviors and women's perception regarding justification of IPV, especially in those women with an unplanned pregnancy, could help to reduce IPV. Further, changing social norms that condone violence through advocacy and awareness

creation might help in preventing partner violence [63, 64]. To support this awareness creation using local evidence, further research that investigates the consequence of IPV on pregnancy outcomes is needed. Law enforcement agencies should strictly implement legislation that prevents violence against women, which in the long term will aid in creating societies that do not tolerate violence [40].

## Acknowledgments

We would like to extend our appreciation to the University of Newcastle, Australia for providing technical and financial support for this study. We would also like to thank all mothers who participated in this study and their commitment to respond to our questions. We would like to thank Ms.Natalia Soeters for language proof reading and editing this manuscript.

## Author Contributions

**Conceptualization:** Abdulbasit Musa, Catherine Chojenta, Deborah Loxton.

**Formal analysis:** Abdulbasit Musa.

**Funding acquisition:** Abdulbasit Musa.

**Investigation:** Abdulbasit Musa.

**Methodology:** Abdulbasit Musa, Catherine Chojenta, Deborah Loxton.

**Project administration:** Abdulbasit Musa.

**Resources:** Abdulbasit Musa, Catherine Chojenta, Deborah Loxton.

**Software:** Abdulbasit Musa.

**Supervision:** Abdulbasit Musa, Catherine Chojenta, Deborah Loxton.

**Validation:** Abdulbasit Musa, Catherine Chojenta, Deborah Loxton.

**Writing – original draft:** Abdulbasit Musa.

**Writing – review & editing:** Abdulbasit Musa, Catherine Chojenta, Deborah Loxton.

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
