## [Decision Letter · Decision Letter 0]

23 Mar 2020

PONE-D-19-32908

High rate of partner violence during pregnancy in eastern Ethiopia: Findings from a facility-based study

PLOS ONE

Dear Mr. Musa,

Thank you for submitting your manuscript to PLOS ONE. After careful consideration, we feel that it has merit but does not fully meet PLOS ONE’s publication criteria as it currently stands. Therefore, we invite you to submit a revised version of the manuscript that addresses the points raised during the review process.

This study addresses a very interesting topic with an undoubted interest. Nevertheless, the reviewers have identified some lacks in the manuscript, some of them being relevant. Authors must revise them carefully. If you consider resubmitting the manuscript to PlosOne, I recommend you making a major revision taking into account the two reviewers’ comments. In this sense, I agree especially with the methodological concerns highlighted by them. They must be revised carefully.

Lastly, I apologize for the delay in the response. The past few weeks have been very difficult.

We would appreciate receiving your revised manuscript by May 07 2020 11:59PM. To enhance the reproducibility of your results, we recommend that if applicable you deposit your laboratory protocols in protocols.io, where a protocol can be assigned its own identifier (DOI) such that it can be cited independently in the future. For instructions see: http://journals.plos.org/plosone/s/submission-guidelines#loc-laboratory-protocols

We look forward to receiving your revised manuscript.

Kind regards,

José J. López-Goñi

Academic Editor

PLOS ONE

Additional Editor Comments (if provided):

This study addresses a very interesting topic with an undoubted interest. Nevertheless, the reviewers have identified some lacks in the manuscript, some of them being relevant. Authors must revise them carefully. If you consider resubmitting the manuscript to PlosOne, I recommend you making a major revision taking into account the two reviewers’ comments. In this sense, I agree especially with the methodological concerns highlighted by them. They must be revised carefully.

Lastly, I apologize for the delay in the response. The past few weeks have been very difficult.

Journal Requirements:

2. Please address the following:

- Please include additional information regarding the survey or questionnaire used in the study and ensure that you have provided sufficient details that others could replicate the analyses. For instance, if you developed a questionnaire as part of this study and it is not under a copyright more restrictive than CC-BY, please include a copy, in both the original language and English, as Supporting Information.

- Please describe how verbal consent was documented and witnessed.

- Please refrain from stating p values as 0.000, either use the exact number or the format p<0.001.

3. Your ethics statement must appear in the Methods section of your manuscript. If your ethics statement is written in any section besides the Methods, please move it to the Methods section and delete it from any other section. Please also ensure that your ethics statement is included in your manuscript, as the ethics section of your online submission will not be published alongside your manuscript.

Reviewers' comments:

Reviewer's Responses to Questions

**Comments to the Author**

1. Is the manuscript technically sound, and do the data support the conclusions?

Reviewer #1: Yes

Reviewer #2: Partly

2. Has the statistical analysis been performed appropriately and rigorously? 

Reviewer #1: Yes

Reviewer #2: No

3. Have the authors made all data underlying the findings in their manuscript fully available?

Reviewer #1: Yes

Reviewer #2: Yes

4. Is the manuscript presented in an intelligible fashion and written in standard English?

Reviewer #1: Yes

Reviewer #2: Yes

5. Review Comments to the Author

Reviewer #1: PONE-D-19-32908 High rate of partner violence during pregnancy in eastern Ethiopia: Findings from a facility-based study.

The study examines the prevalence and associated factors of intimate partner violence (IPV) during pregnancy among women who had given birth in public hospitals in the Harari region of Ethiopia. In addition, it points out some risk factors that could guide the intervention.

The authors justify the need for this study because the limited information on this problem in this specific region. However, they mention the existence of other studies in Ethiopia. Could the authors indicate what differentiates this region from those already been studied? In other words, why study this specific region is relevant?

Abstract

Please, correct the sentence: “Variables with a pvalue of £0.05 were considered to have an association with intimate partner violence during pregnancy”.

Introduction

After indicating that Ethiopia is a multi-cultural state, the authors describe dowry as a practice that increases the risk of IPV for women. They also describe two types of dowry with potentially different results for women. However, it is not clear how this practice is in the Harari region.

Method

It is not clear if data collection was completed before or after births. Hence, it is not obvious what the authors mean when they say “the most recent pregnancy”? Is it the current pregnancy? In page 13, it seems that some interviews were completed before and other after births. Please, clarify this.

It would be necessary to know the format of the questions posed and some examples of them.

Statistical Analysis

This section needs a more detailed description of the steps followed in the analysis. For instance, could you indicate how you set the levels of the outcome to compute the logistic regressions? (i.e., was it used a "yes" or "no" binary category?). Could you indicate the method used to introduce the predictor variables into the analyses? Moreover, it is not clear how and why the authors computed a binary regression and, subsequently, a multiple logistic regression. How did they change an outcome with two levels to more than two levels? The authors need to explain this better?

Discussion

The discussion outlines the findings and situates these alongside the reviewed research. The paper also refers to some study limitations. However, the authors could also refer to the consequences of IPV on the health of women and their children as a means to demand more intensely a change in social norms that condone violence. In a social context with such a high acceptance of IPV, hospitals can do more than point out that violence exists. They can focus on the seriousness of the consequences that violence entails in order to start raising awareness.

For the future, the authors could find interesting to analyze the consequences of IPV on the health of women and neonates.

It is strange that no woman accepted the help offered by the violence she was suffering. Authors should try to explain this.

Reviewer #2: Thank you very much for letting me read and comment this interesting manuscript. The aim of the study was to determine the prevalence of IPV during pregnancy in a region of Ethiopia and to explore IPV related factors. Although this is a widely studied topic throughout the world, one more article will be always necessary. Shown below, I make some comments hoping they will be useful to improve the manuscript.

Generally, authors made an exploratory analysis. I personally think that a hypothesys-based analysis would make more sense. In my opinión, the current approach is a little bit confusing.

Summary:

Please, provide the explanation for AOR and CI.

The second sentence of the conclusion is not necessarily drawn from the results.

Introduction:

The first paragraph does not seem necessary to me.

The paragraph on page 4 that begins with “The relationship between dowry payment…” should be revised. From y point of view, too much importance is given to dowry as a explanatory factor, but then not throughout the text. Perhaps it is only necessary to comment that some factors related to IPV during pregnancy have been identified and give some examples. In the current version, it is not known whether the factors cited are related to IPV or IPV during pregnancy.

Methods:

The first section could be entitled "Study design" and include soon the mention to "cross-sectional study". Ethical statements could be placed in this same section.

The dowry payment exposure was used to calculate the sample size, but it is not known why and, in any case, an estimate of its magnitude is lacked.

How many items does the questionnaire have? How many for each type of violence? Could you provide details of its validity?

Generally, there are too many variables. I believe that a selection could be made with the most relevant according to the scientific literature and the most explanatory according to the preliminary analyses.

Results:

I would prefrer to see the first tables and figures with the total percentages (like now), but also the percentages according to having suffered from IPV (and not). That is, three columns, one with total n (%), one with n (%) for those without IPV, and another column for those with IPV. A fourth column can be displayed with p-values.

Figures mixing absolute and relative values don't make much sense.

In Table 5, mixing the variables that reflect the partner’s control and the attitude towards IPV with the other variables is confusing for me. There may be overlapping issues and it does not help to understand the phenomenon. These two variables could be used to make strata, and study the effect of the other variables on them. In sum, these variables could be on the causal pathaway.

Conclusions:

Some of them are not based on results. Some are general comments, probably true, but not followed directly from this study. From this work it can be concluded, for instance, that a pregnancy monitoring program is necessary, which can detect partner’s control behaviors and women justification of IPV, especially in those women with an unplanned pregnancy.

6. PLOS authors have the option to publish the peer review history of their article (what does this mean?). If published, this will include your full peer review and any attached files.

Reviewer #1: Yes: Rosaura Gonzalez-Mendez

Reviewer #2: No

---

## [Author Response · Author response to Decision Letter 0]

15 Apr 2020

Response to Editor and reviewers

Response to Editor

Comment 1: Please include additional information regarding the survey or questionnaire used in the study and ensure that you have provided sufficient details that others could replicate the analyses. For instance, if you developed a questionnaire as part of this study and it is not under a copyright more restrictive than CC-BY, please include a copy, in both the original language and English, as Supporting Information

Response 1: The questionnaire we used to collect this data was obtained from WHO Multi-country VAW study. This questionnaire will be submitted and published in University Newcastle online thesis repository as part of PhD thesis. However, we extracted specific items that were used to measure the outcome variable and included them in the methodology section in the current manuscript as a table (Table 1). 

Comment 2: Please describe how verbal consent was documented and witnessed

Response 2: To address this, the following sentences were added to information under Ethical clearance “First, the interviewer read out participant’s information sheet slowly. Information regarding purpose of the study, risk and benefit of participating, measure taken to protect participants’ privacy and their choice to participate in the study were explained. Then, they were given adequate time to ask questions to ensure they had a clear understanding of the study. After their questions were addressed, their consent was sought by asking the participant to respond affirmatively to a series of statements. The interviewer signed and dated the consent form”.

Comment 3: Please refrain from stating p values as 0.000, either use the exact number or the format p<0.001.

Response 3:Comment accepted and corrected in the current manuscript.

comment 4: Your ethics statement must appear in the Methods section of your manuscript. If your ethics statement is written in any section besides the Methods, please move it to the Methods section and delete it from any other section. Please also ensure that your ethics statement is included in your manuscript, as the ethics section of your online submission will not be published alongside your manuscript

Response 4: Our ethical clearance statement appears under the methodology section. In the current manuscript version, the approval numbers for respective Ethical clearance have been inserted

Response to reviewer 1

Comment 1

General comments

The authors justify the need for this study because the limited information on this problem in this specific region. However, they mention the existence of other studies in Ethiopia. Could the authors indicate what differentiates this region from those already been studied? In other words, why study this specific region is relevant?

Response 1: This region is different from most of the regions in Ethiopia. Some of these differences are described below:

A) Residency: The Ethiopian population is predominantly rural (86%). However, Harari region is the only region in Ethiopia where the majority of its population live in urban locations (55%) as described under the study setting in the manuscript. Residency is one of the important factors that can affect IPV(Naved and Persson 2008).

B) Religion: about one third of the Ethiopian population are Muslim but in Harari region more than two thirds of the population are Muslim. Differences in religion might contribute to IPV as the religion might be misused to justify IPV. For example, in some Muslim countries, selective quotes from the Quran may be used to ‘prove’ that it is allowable for a man to ‘beat’ his wife (Douki et al. 2003) which might embolden the perpetrator

C) Substance use: Although about 16% of males in Ethiopia use Khat, in Eastern Ethiopia (including Harari region) a relatively higher proportion of the population use Khat, Evidence has indicated more than 32% of women and 73% of males use Khat (CSA [Ethiopia] and ICF 2016). Using Khat and other drugs were associated with an increased risk of IPV (Kassa and Menale 2016; Lencha et al. 2019). In the Ethiopian context, regions are semi-autonomous and they have their own health bureau that leads interventions related to health in the region. Therefore, region based information regarding IPV is crucial to make intervention that reduce IPV among pregnant women in the region. 

This detail is included in the introduction part of the manuscript as follows ”Harari region has the highest proportion of population who reside in urban (55%) than any other regions in Ethiopia and the majority of its population are Muslim (69%) (Central Statistical Agency 2013). Both residency (Naved and Persson 2008) and religion (Douki et al. 2003) are important factors that can affect the prevalence of IPV. For example, evidence indicated that in some Muslim countries, selective quotes from the Quran may be used to ‘prove’ that it is allowable for a man to ‘beat’ his wife (Douki et al. 2003) indicating the role of religion in promoting social control. Therefore, the generalizability of the study that was conducted in one part of the country might not provide the true prevalence of IPV in the other region as IPV might be affected by within-country differences in cultures and societal norms (Linos et al. 2013) since Ethiopia is a multi-cultural state”

Comment 2: Abstract

Please, correct the sentence: “Variables with a p-value of ≤0.05 were considered to have an association with intimate partner violence during pregnancy

Response 2: This sentence is corrected as follows “Variables with a p-value of ≤0.05 were considered to have a significant association with intimate partner violence during pregnancy”

Comment 3; Introduction

After indicating that Ethiopia is a multi-cultural state, the authors describe dowry as a practice that increases the risk of IPV for women. They also describe two types of dowry with potentially different results for women. However, it is not clear how this practice is in the Harari region

Response 3: The dowry practice in Harari region is similar to Ethiopia. In Harari region, the dowry is given to the bride’s family as cash or cattle. This statement is included in the current version of the manuscript as follows: “The relationship between dowry payment and partner violence has long been established (Batra 1988). Unlike countries in Asia that offer a dowry to the groom’s family (Banerjee 2013), in Ethiopia a dowry of cash or cattle is given to the bride’s family before the marriage (Regassa, Mitiku, and Hailu 2019); this is the practice in Harari region.”

Comment 4: Method

It is not clear if data collection was completed before or after births. Hence, it is not obvious what the authors mean when they say “the most recent pregnancy”? Is it the current pregnancy? In page 13, it seems that some interviews were completed before and other after births. Please, clarify this.

Response 4: Under the participant section, we mentioned, “All women who gave birth in the public hospitals found in the Harari region were included in the eligible study population”. This is to indicate that data collection was conducted on women who gave birth in the hospitals (during postpartum period), not on pregnant women. To avoid confusion, under the “participants” sub section we added this sentences “Participants were recruited following the birth of their baby in the hospitals”. Further, we have replaced the “current pregnancy” with the “most recent pregnancy” in the current manuscript.” We used the most recent pregnancy to indicate that women were asked whether their experienced IPV during their last pregnancy. 

Comment 5: It would be necessary to know the format of the questions posed and some examples of them

Response 5: This study is part of larger PhD study to investigate IPV in the region. Although the questionnaire has multiple sections that will be submitted to the University for online publication after the completion of the study, we extracted the part that addressed IPV and summarized this in the table in the methodology section. The questionnaire we have used is adapted for the Ethiopian context, as Ethiopia is one of the countries that was included in WHO multi country study that was conducted in 2005. Therefore, we used the same tool that are commonly being used in Ethiopia by researchers (Mohammed et al. 2017; Yimer et al. 2014; Abebe Abate, Admassu Wossen, and Tilahun Degfie 2016).

Comment 6: Statistical Analysis

This section needs a more detailed description of the steps followed in the analysis. For instance, could you indicate how you set the levels of the outcome to compute the logistic regressions? (i.e., was it used a "yes" or "no" binary category?). Could you indicate the method used to introduce the predictor variables into the analyses? Moreover, it is not clear how and why the authors computed a binary regression and, subsequently, a multiple logistic regression. How did they change an outcome with two levels to more than two levels? The authors need to explain this better?

Response 6: Comment accepted and the following modification was made. 

The detail about items used to measure each aspect of IPV was provided in the manuscript under methodology by using table. As it appears on the Table 1, physical IPV was assessed by 6 items, psychological IPV was assessed by 4 items while sexual IPV was assessed by 3 items. All the questions asked were ‘Yes’ or ‘No’ question. Women who reported ‘Yes” for at least one of the physical IPV items were considered to have experienced physical IPV. Similarly, women who responded ‘Yes’ for at least one of the emotional IPV measurement items were considered to have experienced emotional IPV while those who replied “yes’ for at least one of the three sexual IPV items were considered to have Sexual IPV. For overall IPV, women who reported experiencing any types of IPV (whether physical, emotional or sexual) were considered to have IPV during the most recent pregnancy. This overall IPV was used as an outcome and it is categorical variable with only two possible outcomes (Yes and No for IPV).

Regarding analytical statistics, since the outcome variable is a categorical variable we can do both binary logistic regression and multiple logistic regression. In binary logistic regression, we entered variables one by one in the logistic regression model to see its individual associations. However, to conduct multiple logistic regression, we entered more than one variable to get the adjusted relationship between IPV and other explanatory variables. The predictor variables were selected based on the findings of relevant literature as well as statistical significance.

Comment 7: Discussion

The discussion outlines the findings and situates these alongside the reviewed research. The paper also refers to some study limitations. However, the authors could also refer to the consequences of IPV on the health of women and their children as a means to demand more intensely a change in social norms that condone violence. In a social context with such a high acceptance of IPV, hospitals can do more than point out that violence exists. They can focus on the seriousness of the consequences that violence entails in order to start raising awareness. For the future, the authors could find interesting to analyze the consequences of IPV on the health of women and neonates. It is strange that no woman accepted the help offered by the violence she was suffering. Authors should try to explain this

Response 7: This is very important point and we made modifications to address these comments. We agree that hospitals could do more than indicating that violence exists. However, to support awareness creation it is important to know the impact of IPV on adverse pregnancy outcomes using local studies. Therefore, we included recommendations that indicate the importance conducting studies to investigate the consequence of IPV. This is described in the conclusion as follows; “….to support this awareness creation using local evidence, further research that investigate the consequences of IPV on pregnancy outcomes in the Ethiopian context are needed” As mentioned earlier, this is study is part of larger study that include investigating the consequence of IPV. We have aim that investigated the consequence of IPV and the manuscript that was written on the consequence of IPV (adverse pregnancy outcomes) is currently under review. In that manuscript, broader finding-based recommendation that are critically important to address IPV were included. 

Furthermore, the explanation regarding why woman do not accept offers was described in relation to attitudes towards IPV in the discussion. It was presented as follows: “… It is important to note that none of the women who experienced IPV in the current study took up the offer of support. Although further study is needed to understand why abused women may not want to get support, this might indicate an overall tolerance of violence within the community and its role in fostering a conducive environment for abusers (Kurz 1989)”.

Response to Reviewer 2

Comment 1: General comment

Authors made an exploratory analysis. I personally think that a hypothesis-based analysis would make more sense. In my opinion, the current approach is a little bit confusing.

Response 1: Although we clearly did not describe all potential variables that were investigated in relation to IPV in the form of a null hypothesis and an alternative hypothesis, our analysis still tested the significance of associations between variables. As was described in Methodology section, we have used previous literature to identify variables potentially associated with IPV and then testing those associations. 

In the current manuscript, to clearly indicate this , we included the research question as follows “In other word, the study attempted to address the following question: ‘is there an association between IPV and a given set of sociodemographic factors, marital and behavioural factors, reproductive and physical health-related factors, attitude towards IPV and partner controlling behaviour?.

Comment 2: Abstract

Please, provide the explanation for AOR and CI.

Response 2: We have put the explanation for both AOR and CI as a footnote in Table 6 in the current version of the manuscript.

Comment 3: The second sentence of the conclusion is not necessarily drawn from the results.

Response 3: We corrected the conclusion as per the reviewer’s recommendation as follows: “The prevalence of intimate partner violence during pregnancy was found to be high. Pregnancy monitoring programs, which can detect and intervene with regard to partner’s controlling behaviors and women’s perception regarding justification of IPV, especially in those women with an unplanned pregnancy, could help to reduce IPV. Further, changing social norms that condone violence through advocacy and awareness creation might help in preventing partner violence”

Comment 4: Introduction:

The first paragraph does not seem necessary to me.

Response 4: The first paragraph talks about the definition of IPV used in the current manuscript. We felt that this is important to introduce the reader with the concept of IPV, how it was defined and which definition is used in this manuscript. We have therefore opted to leave the paragraph. 

Comment 5: The paragraph on page 4 that begins with “The relationship between dowry payment…” should be revised. From my point of view, too much importance is given to dowry as a explanatory factor, but then not throughout the text. Perhaps it is only necessary to comment that some factors related to IPV during pregnancy have been identified and give some examples. In the current version, it is not known whether the factors cited are related to IPV or IPV during pregnancy

Response 5: It is true that although we mentioned other factors that are associated with IPV, we gave more emphasis to dowry practices. And as per Reviewer 1, comment, we have clarified the importance of dowry in Ethiopia. We did this to introduce the reader to dowry practices in the Ethiopian context because we have used this variable for calculating sample size. 

Comment 6: Methods:

The first section could be entitled "Study design" and include soon the mention to "cross-sectional study".

Response 6: Comment accepted and corrected in the current version as follows: “Study design: cross-sectional survey of women attending public hospitals found in the Harari region was conducted from 25 November 2018 to 28 April 2019”.

Comment 7: The dowry payment exposure was used to calculate the sample size, but it is not known why and, in any case, an estimate of its magnitude is lacked.

Response 7: This study is part of larger study that investigated the magnitude, associated factors, attitudes and outcomes of IPV during pregnancy in eastern Ethiopia. The sample size that addresses all aims was calculated by using two scenarios (the single proportion population formula and associated factors). Obtained from previous literature, the prevalence of IPV, and proportion of women with attitude supportive of IPV were used to calculate sample size and the largest sample size (from the two proportion) was identified. Similarly, for associated factors, we have used variable that gave us larger sample size considering associated factors of IPV, factors associated with having a perception that justify IPV and pregnancy outcomes. From overall sample size calculated including sample size calculated by using single population proportion formula, the sample size calculated using dowry practice gave us largest sample size. Therefore we used it for calculating overall sample size. As we never used the proportion of dowry practices to calculate sample size, we do not think that it is important to mention the prevalence of dowry practices in this method. That is why we described dowry practice as a factor rather than describing its proportion. 

Comment 8: How many items does the questionnaire have? How many for each type of violence? Could you provide details of its validity?

Response 8: Overall, there are 13 items used to assess IPV. Physical IPV was assessed by 6 items, psychological IPV was measured by 4 items while sexual IPV was measured by 3 items. This questionnaire was adapted from WHO multi-country study questionnaire. WHO conducted study to provide the overall global prevalence of violence against women in 2005, and Ethiopia was one of the countries that was included in this global study. The overall questionnaire was adapted to the Ethiopian context by removing some of the items that could not be applied in Ethiopia context, as described in the Methodology section of WHO study. These items included questions that were designed to addresses the impact of violence on mental health. However, specific items that measured IPV were found to be valid in the Ethiopian context and none of them were removed. This questionnaire has been widely used by different researchers in Ethiopia to study IPV (Mohammed et al. 2017; Yimer et al. 2014; Abebe Abate, Admassu Wossen, and Tilahun Degfie 2016).

For clarity, we have included the items used to measure IPV in the manuscript in Methodology (Table 1).

Comment 9: Generally, there are too many variables. I believe that a selection could be made with the most relevant according to the scientific literature and the most explanatory according to the preliminary analyses.

Response 9: We agree with the reviewer. As mentioned previously, we have collected many variables that are intended to address all outcomes we were looking to address in the wider study. Although some of them can provide further descriptions about the population, it is not necessary to include all of them in the current manuscript. Therefore, we removed some of the details that were addressed in other manuscripts. For example, we removed some of the variables that are related to pregnancy outcome as we have another manuscript dedicated to this. 

Specific to the variables entered in to the multiple logistic regression, first we did not included all variables in the binary logistic regression. It is true that model with more variables presents less statistical power, which can lead to wrong conclusion. To avoid this, we first fitted binary logistic regression using relevant variables that were only supported by previous literature. Then, we observed its level of significant in bivariate analysis and variable with a p-value of ≤0.05 were transported to the multiple logistic regression. In addition, there were variables that were significant but were not transferred to multiple logistic regression (E.g. gravidity). This decision was made because parity and gravidity are similar concepts and the variables were highly correlated. Therefore, we excluded gravidity from multivariate analysis. Overall, we have used a combination of statistical significance and relevant literature to determine variables that were entered into the logistic regression model.

To address this, the methodology section of the manuscript was edited as follow: “three logistic regression models were fitted to investigate the associated factors of IPV. In the first model (Model 1), binary logistic regression was conducted to determine associations between IPV and potential explanatory variables. All potential explanatory variables that were fitted in binary logistic regression were obtained from relevant literature. In the second model (Model 2), variables (excluding attitude towards IPV and partner controlling behaviour) that demonstrated a significant association (with a p-value of �0.05) in bivariate analysis were exported to a multiple logistic regression model for multivariate analysis”.

Comment 10: Results:

I would prefer to see the first tables and figures with the total percentages (like now), but also the percentages according to having suffered from IPV (and not). That is, three columns, one with total n (%), one with n (%) for those without IPV, and another column for those with IPV. A fourth column can be displayed with p-values.

Response 10: Comment accepted and appropriate changes made.

Comment 11: Figures mixing absolute and relative values don't make much sense

Response 11: Comment accepted and corrected. The figures were described by percentages only

Comment 12: In Table 5, mixing the variables that reflect the partner’s control and the attitude towards IPV with the other variables is confusing for me. There may be overlapping issues and it does not help to understand the phenomenon. These two variables could be used to make strata, and study the effect of the other variables on them. In sum, these variables could be on the causal pathway.

Response 12: Comment accepted and modified as per reviewer’s recommendation.

In this study, we were not interested in conducting pathway analysis. Rather, we included partner control and women’s attitudes towards IPV with other variables considering that all variables can independently contribute for IPV. This approach was used in a number of previous studies (Black, E., Worth, H., Clarke, S. et al. Prevalence and correlates of intimate partner violence against women in conflict affected northern Uganda: a cross-sectional study. Confl Health 13, 35 (2019). https://doi.org/10.1186/s13031-019-0219-8, Tlapek, S. M. (2015) ‘Women’s Status and Intimate Partner Violence in the Democratic Republic of Congo’, Journal of Interpersonal Violence, 30(14), pp. 2526–2540. doi: 10.1177/0886260514553118, Shai N, Pradhan GD, Chirwa E, Shrestha R, Adhikari A, Kerr-Wilson A (2019) Factors associated with IPV victimisation of women and perpetration by men in migrant communities of Nepal. PLoS ONE 14(7): e0210258. https://doi.org/10.1371/journal.pone.0210258). However, in the current version of the manuscript, we constructed three different models. The first model contains bivariate analysis of all variables, while the second model contain all variables found to have significant associations in bivariate analysis, excluding attitude towards IPV and controlling behaviour. The last (third) model included all variables found to have significant associations, including attitude towards IPV and partner controlling behaviour. Within both multivariate models, there was no big change in terms of effect size though duration of marriage that had become insignificant in model 2 became significant in Model 3. To investigate whether attitude towards IPV and partner controlling behaviour moderates the duration of marriage, the interaction analysis was done but it was not significant. Therefore, we did not include this interaction term in the model, as it was not significant. The discussion was made based on the final model (Model 3).

Comment 13 : Some of them are not based on results. Some are general comments, probably true, but not followed directly from this study. From this work it can be concluded, for instance, that a pregnancy monitoring program is necessary, which can detect partner’s control behaviors and women justification of IPV, especially in those women with an unplanned pregnancy.

Response 13: Comment accepted and the conclusion is modified based on reviewer’s suggestion. The conclusion was re-stated as follows: “The prevalence of IPV during pregnancy is high; two in five women experienced IPV during their most recent pregnancy. Pregnancy monitoring programs, which can detect and intervene with regard to partner’s controlling behaviors and women’s perception regarding justification of IPV, especially in those women with an unplanned pregnancy, could help to reduce IPV. Further, changing social norms that condone violence through advocacy and awareness creation might help in preventing partner violence (Wagman et al. 2018; Rivas et al. 2015). To support this awareness creation using local evidence, further researches that investigate the consequence of IPV on pregnancy outcomes are needed. Law enforcement agencies should strictly implement legislation that prevents violence against women, which in the long term will aid in creating societies that do not tolerate violence (World Health Organization 2010).

We have used these references to justify our responses.

References 

Abebe Abate, B., B. Admassu Wossen, and T. Tilahun Degfie. 2016. 'Determinants of intimate partner violence during pregnancy among married women in Abay Chomen district, Western Ethiopia: a community based cross sectional study', BMC women's health, 16: 16.

Banerjee, Priya R. 2013. 'Dowry in 21st-Century India: The Sociocultural Face of Exploitation', Trauma, Violence, & Abuse, 15: 34-40.

Batra, N. D. 1988. 'Why this growing atrocities on women?', Yojana, 32: 32-4.

Central Statistical Agency. 2013. "Population Projection of Ethiopia for All Regions At Wereda Level from 2014 – 2017." In, edited by Federal Demographic Republic of Ethiopia Central Statistical Agency. Addis Ababa: Central Statistical Agency 

CSA [Ethiopia], and ICF. 2016. " Ethiopia Demographic and Health Survey 2016. , ." In. Addis Ababa, Ethiopia and Rockville, Maryland, USA.

Douki, S., F. Nacef, A. Belhadj, A. Bouasker, and R. Ghachem. 2003. 'Violence against women in Arab and Islamic countries', Arch Womens Ment Health, 6: 165-71.

Kassa, Zemenu Yohannes, and Alemu Workineh Menale. 2016. 'Physical violence and associated factors during pregnancy in Yirgalem town, South Ethiopia', Current Pediatric Research, 20: 37-42.

Kurz, Demie. 1989. 'SOCIAL SCIENCE PERSPECTIVES ON WIFE ABUSE:: Current Debates and Future Directions', Gender & Society, 3: 489-505.

Lencha, B., G. Ameya, G. Baresa, Z. Minda, and G. Ganfure. 2019. 'Intimate partner violence and its associated factors among pregnant women in Bale Zone, Southeast Ethiopia: A cross-sectional study', PLoS One, 14: e0214962.

Linos, Natalia, Natalie Slopen, S. V. Subramanian, Lisa Berkman, and Ichiro Kawachi. 2013. 'Influence of Community Social Norms on Spousal Violence: A Population-Based Multilevel Study of Nigerian Women', American Journal of Public Health, 103: 148-55.

Mohammed, B. H., J. M. Johnston, J. I. Harwell, H. Yi, K. W. K. Tsang, and J. A. Haidar. 2017. 'Intimate partner violence and utilization of maternal health care services in Addis Ababa, Ethiopia', BMC Health Serv Res, 17.

Naved, R. T., and L. A. Persson. 2008. 'Factors associated with physical spousal abuse of women during pregnancy in Bangladesh', Int Fam Plan Perspect, 34: 71-8.

Regassa, Megersa, Terefe Mitiku, and Waktole Hailu. 2019. 'Addooyyee: Girl’s Indigenous Friendship Institution in Oromoo, Ethiopia', International Journal of Multicultural and Multireligious Understanding, 6.

Rivas, C., J. Ramsay, L. Sadowski, L. L. Davidson, D. Dunne, S. Eldridge, K. Hegarty, A. Taft, and G. Feder. 2015. 'Advocacy interventions to reduce or eliminate violence and promote the physical and psychosocial well-being of women who experience intimate partner abuse', Cochrane Database Syst Rev: Cd005043.

Wagman, J. A., R. H. Gray, N. Nakyanjo, K. A. McClendon, E. Bonnevie, F. Namatovu, G. Kigozi, J. Kagaayi, M. J. Wawer, and F. Nalugoda. 2018. 'Process evaluation of the SHARE intervention for preventing intimate partner violence and HIV infection in Rakai, Uganda', Eval Program Plann, 67: 129-37.

World Health Organization. 2010. "Violence prevention: The Evidence." In. Geneva: World Health Organisatioon.

Yimer, Tenaw, Tesfaye Gobena, Gudina Egata, and Habtamu Mellie. 2014. 'Magnitude of Domestic Violence and Associated Factors among Pregnant Women in Hulet Ejju Enessie District, Northwest Ethiopia', Advances in Public Health, 2014: 8.

---

## [Decision Letter · Decision Letter 1]

12 May 2020

PONE-D-19-32908R1

High rate of partner violence during pregnancy in eastern Ethiopia: Findings from a facility-based study

PLOS ONE

Dear Mr. Musa,

Thank you for submitting your manuscript to PLOS ONE. After careful consideration, we feel that it has merit but does not fully meet PLOS ONE’s publication criteria as it currently stands. Therefore, we invite you to submit a revised version of the manuscript that addresses the points raised during the review process.

Thank you very much for your work and effort. As the reviewers have said, the main concerns have been addressed. In this round of review, the reviewer 2 has made two comments. Please consider them because they could improve the final version of the manuscript.

We would appreciate receiving your revised manuscript by Jun 26 2020 11:59PM. To enhance the reproducibility of your results, we recommend that if applicable you deposit your laboratory protocols in protocols.io, where a protocol can be assigned its own identifier (DOI) such that it can be cited independently in the future. For instructions see: http://journals.plos.org/plosone/s/submission-guidelines#loc-laboratory-protocols

We look forward to receiving your revised manuscript.

Kind regards,

José J. López-Goñi

Academic Editor

PLOS ONE

Additional Editor Comments (if provided):

Thank you very much for your work and effort. As the reviewers have said, the main concerns have been addressed. In this round of review, the reviewer 2 has made two comments. Please consider them because they could improve the final version of the manuscript.

Reviewers' comments:

Reviewer's Responses to Questions

**Comments to the Author**

1. If the authors have adequately addressed your comments raised in a previous round of review and you feel that this manuscript is now acceptable for publication, you may indicate that here to bypass the “Comments to the Author” section, enter your conflict of interest statement in the “Confidential to Editor” section, and submit your "Accept" recommendation.

Reviewer #1: All comments have been addressed

Reviewer #2: All comments have been addressed

2. Is the manuscript technically sound, and do the data support the conclusions?

Reviewer #1: Yes

Reviewer #2: Yes

3. Has the statistical analysis been performed appropriately and rigorously? 

Reviewer #1: Yes

Reviewer #2: Yes

4. Have the authors made all data underlying the findings in their manuscript fully available?

Reviewer #1: Yes

Reviewer #2: Yes

5. Is the manuscript presented in an intelligible fashion and written in standard English?

Reviewer #1: Yes

Reviewer #2: Yes

6. Review Comments to the Author

Reviewer #1: (No Response)

Reviewer #2: The authors addressed my primary concerns, thank you. Now, I only have two comments:

Thank you for putting the explanation for both AOR and CI as a footnote in Table 6, but I think that it also be helpful in the abstract.

Thank you very much for the explanation regardign sample size calculation. I sugges to add some of your comments. For expample: The sample size was calculated using significant factors of partner violence during pregnancy obtained from a previous study conducted in Ethiopia [16]. Dowry payment maximized sample size, therefore we used it as exposure variable for calculations….

7. PLOS authors have the option to publish the peer review history of their article (what does this mean?). If published, this will include your full peer review and any attached files.

Reviewer #1: Yes: Rosaura Gonzalez-Mendez

Reviewer #2: No

---

## [Author Response · Author response to Decision Letter 1]

12 May 2020

Dear Editors

We appreciate your effort in considering our article for further review. We also thank the reviewers for taking time to review the manuscript again. We have made the required changes to the manuscript and have summarized our responses to Reviewr-2 as follow:

Comments and Responses (Reviewer 2)

 Comment 1: Thank you for putting the explanation for both AOR and CI as a footnote in Table 6, but I think that it also be helpful in the abstract

Response 1: Comment accepted, both AOR and CI were written in full statement in the current manuscript abstract

Comment 2:I suggest to add some of your comments. For example: The sample size was calculated using significant factors of partner violence during pregnancy obtained from a previous study conducted in Ethiopia [16]. Dowry payment maximized sample size, therefore we used it as exposure variable for calculations…

Response 2: Comment accepted. The section is modified as follows: The sample size was calculated using significant factors of partner violence during pregnancy obtained from a previous study conducted in Ethiopia [16]. Dowry payment maximized sample size; therefore, we used it as the exposure variable for calculating sample size considering the following parameters: dowry payment as the exposure variable, power level of 80%, and the ratio of unexposed to exposed of 1.0….

---

## [Editor Report · Decision Letter 2]

15 May 2020

High rate of partner violence during pregnancy in eastern Ethiopia: Findings from a facility-based study

PONE-D-19-32908R2

Dear Dr. Musa,

We are pleased to inform you that your manuscript has been judged scientifically suitable for publication and will be formally accepted for publication once it complies with all outstanding technical requirements.

With kind regards,

José J. López-Goñi

Academic Editor

PLOS ONE
---

## [Editor Report · Acceptance letter]

19 May 2020

PONE-D-19-32908R2 

High rate of partner violence during pregnancy in eastern Ethiopia: Findings from a facility-based study 

Dear Dr. Musa:

I am pleased to inform you that your manuscript has been deemed suitable for publication in PLOS ONE. Congratulations! Your manuscript is now with our production department. 

With kind regards,

on behalf of

Dr. José J. López-Goñi 

Academic Editor

PLOS ONE